# Dynamics of IgM and IgA Antibody Response Profile Against *Vibrio cholerae* Toxins A, B, and P

**DOI:** 10.3390/ijms26083507

**Published:** 2025-04-09

**Authors:** Salvatore Giovanni De-Simone, Paloma Napoleão-Pêgo, Guilherme Curty Lechuga, Joao Pedro Rangel Silva Carvalho, Sergian Vianna Cardozo, Alexandre Oliveira Saisse, Carlos Medicis Morel, David William Provance, Flavio Rocha da Silva

**Affiliations:** 1Center for Technological Development in Health (CDTS)/National Institute of Science and Technology for Innovation in Neglected Population Diseases (INCT-IDPN), Oswaldo Cruz Foundation (FIOCRUZ), Rio de Janeiro 21040-900, Brazil; paloma.pego@fiocruz.br (P.N.-P.); guilherme.curty@fiocruz.br (G.C.L.); joaoprsc@id.uff.br (J.P.R.S.C.); pbrava5@gmail.com (A.O.S.); carlos.morel@fiocruz.br (C.M.M.); bill.provance@fiocruz.br (D.W.P.J.); flaviorochafiocruz@gmail.com (F.R.d.S.); 2Epidemiology and Molecular Systematics Laboratory (LEMS), Oswaldo Cruz Institute, Oswaldo Cruz Foundation (FIOCRUZ), Rio de Janeiro 21040-900, Brazil; 3Program of Post-Graduation on Science and Biotechnology, Department of Molecular and Cellular Biology, Biology Institute, Federal Fluminense University, Niteroi 22040-036, Brazil; 4Program of Post-Graduation on Parasitic Biology, Oswaldo Cruz Institute, Oswaldo Cruz Foundation, Rio de Janeiro 21040-900, Brazil; 5Department of Health, Graduate Program in Translational Biomedicine (BIOTRANS), University of Grande Rio (UNIGRANRIO), Caxias 25071-202, Brazil; sergian.cardozo@unigranrio.edu.br

**Keywords:** cholera toxin, IgA epitope, IgM epitopes, peptide microarray, synthetic peptide, MAPs, ELISA-peptide

## Abstract

The first immune response controls many bacterial and viral inflammatory diseases. Oral immunization with cholera toxin (CT) elicits antibodies and can prevent cholerae in endemic environments. While the IgG immune response to the toxin is well-documented, the IgA and IgM epitopes responsible for the initial immune reaction to the toxin remained uncharted. In this study, our objective was to identify and characterize immunologically and structurally these IgA and IgM epitopes. We conducted SPOT synthesis to create two libraries, each containing one hundred twenty-two 15-mer peptides, encompassing the entire sequence of the three chains of the CT protein. We could map continuous IgA and IgM epitopes by testing these membrane-bound peptides with sera from mice immunized with an oral vaccine (Schankol™). Our approach involved topological studies, peptide synthesis, and the development of an ELISA. We successfully identified seven IgA epitopes, two in CTA, two in CTB, and three in protein P. Additionally, we discovered eleven IgM epitopes, all situated within CTA. Three IgA-specific and three IgM-specific epitopes were synthesized as MAP4 and validated using ELISA. We then used two chimeric 45-mer peptides, which included these six epitopes, to coat ELISA plates and screened them with sera from immunized mice. This yielded sensitivities and specificities of 100%. Our findings have unveiled a significant collection of IgA and IgM-specific peptide epitopes from cholera toxins A, B, and P. These epitopes, along with those IgG previously identified by our group, reflect the immunoreactivity associated with the dynamic of the immunoglobulins switching associated with the cholera toxin vaccination.

## 1. Introduction

The Gram-negative bacterium *Vibrio cholerae* is the causative agent of cholera, a life-threatening diarrheal disease primarily transmitted through contaminated food or water. This pathogen continues to pose a global threat, predominantly affecting developing nations with inadequate sewage treatment and clean water infrastructure, notably in Southeast Asia, Latin America, and parts of Africa. Annually, cholera leads to 1.3–4 million cases and 20,000–140,000 cholera-related fatalities worldwide, with nearly half of the victims being children under the age of five [1,2]. Furthermore, the expanding geographical reach of *V. cholerae* due to climate change and rising temperatures underscores the pressing need for further research to enhance our understanding of this pathogen [3,4].

*V. cholerae* strains responsible for disease outbreaks and pandemics fall under the O1 or O139 serogroups, while most *V. cholerae* are considered “non-O1/O139” environmental strains, which may or may not cause gastroenteritis [5,6]. Cholerae is highly contagious and is characterized by severe watery diarrhea, often leading to extreme dehydration, shock, and even death [6]. According to World Health Organization (WHO) reports, more than 80% of cholerae patients can be effectively treated with oral rehydration therapy [7].

The cholerae toxin (CT) is an 85 kDa protein in the AB class of bacterial toxins. It comprises the A subunit (CTA), a 28 kDa heterodimer, and the B subunit (CTB5), with a molecular weight of 55 kDa. CT is transported via cell receptors and converts ATP into cyclic AMP while transferring ADP-ribose from Nicotinamide Adenine Dinucleotide (NAD) to a specific arginine residue in the target protein (Gsa) [8]. This sequential process initiates adenyl cyclase activation, ultimately leading to increased intracellular cAMP levels, which, in turn, triggers the activation of cAMP-dependent protein kinase. This cascade results in protein phosphorylation, altered ion transport, and diarrhea with significant fluid loss [9]. Consequently, public health interventions are of utmost importance to limit the spread of cholera, particularly in regions lacking proper sanitation and clean water infrastructure, where outbreaks can easily occur.

Immunization through cholera vaccines is pivotal in controlling cholerae epidemics and pandemics. Key virulence factors of *V. cholerae*, including CT [10,11], toxin-coregulated pili [12], lipopolysaccharide (LPS) [13], and outer membrane proteins (Omps) [14,15,16], are considered ideal candidates for the development of cholera vaccines. Currently, two types of oral vaccines are in use: attenuated oral cholera vaccines (aOCVs) and inactivated oral cholera vaccines (kOCVs). aOCVs are composed of attenuated whole *V. cholerae* cells and have demonstrated direct effectiveness in protecting populations in endemic regions [17,18]. They have also been deployed in ‘reactive’ vaccination campaigns during outbreaks in non-endemic areas to contain the spread of cholerae [19,20]. In contrast, kOCVs provide herd protection for at least three years and offer short-term immunity with a single dose, which is valuable for outbreak management. However, their optimal efficacy necessitates two doses administered 14 days apart, and these vaccines require refrigeration [21]. These characteristics may limit the ability of kOCVs to swiftly control ongoing outbreaks in unstable or resource-limited settings. Nevertheless, single-dose live aOCVs have demonstrated satisfactory efficacy in challenge studies [22] and early-phase clinical trials in endemic regions [23]. Utilizing live aOCVs in reactive vaccination initiatives may be the most effective approach for reducing cholerae incidence during outbreaks [24,25].

While available oral-killed cholera vaccines are a valuable addition to control efforts, they may offer limited long-term protection, particularly in individuals with no prior exposure to the pathogen, such as children under five years of age, who bear a significant proportion of the global cholerae burden [26]. In contrast, cholerae survivors, including young children, develop high-level protective immunity that lasts three to four years [7,27].

Oral cholera vaccination effectively prevents outbreaks in high-risk areas and manages cholerae in endemic regions [18,28]. However, these vaccines may provide only limited and temporary protection, especially in individuals lacking prior exposure to the pathogen, such as children under five, who represent a significant portion of the global cholerae-affected population [29]. In contrast, cholerae survivors, including young children, develop high-level protective immunity that persists for years [30].

As the supply and utilization of aOCVs expand, crucial questions must be addressed regarding coverage and the vaccination schedule required to establish herd immunity [31]. Additionally, mucosal vaccines, such as nasal vaccines, can activate immune cells located in the mucosal tissues of the upper and lower respiratory tract. This dual immune system stimulation, combined with needle-free administration, supports the development of novel nasal vaccines designed to provide a strong intestinal IgA antibody response and long-lasting immunological memory [32].

Previously, we characterized the secondary immune response following oral cholera vaccination, identifying eighteen IgG B linear epitopes, with six cross-reacting with several bacteria [33]. This study focuses on the primary IgA and IgM immune response epitopes induced by aOCVs with CT. Serological diagnostic assays can be used in large population studies to determine the duration of the immune response, which can potentially be enhanced by identifying CT epitopes recognized by vaccine-induced antibodies. This advancement could improve our understanding of the duration of the immune response and herd protection.

Efficient high-content screening for epitope mapping involves screening peptide libraries representing target coding sequences [34,35]. By synthesizing peptides directly onto cellulose membranes, it is possible to create refined maps of the entire epitome and numerous epitopes within a large protein [36]. SPOT synthesis analysis was employed in this study to identify the IgA and IgM epitopes in three toxins from *V. cholerae* for the development of fast new specific diagnostic assays. Eighteen epitopes (seven IgA and 11 IgM) distinct from the previously identified IgG [33] were identified. ELISA selected and evaluated three of these as multiepitope peptides using sera from mice orally vaccinated.

## 2. Results

### 2.1. Identification of the Immunodominant IgA and IgM Epitopes in Cholera Toxin Subunits

In this section, we aimed to identify the immunodominant epitopes within the three components of the CT, namely CTA (258 aa), CTB (124 aa), and CTP (221 aa). To achieve this, we synthesized two libraries of peptides and assessed their recognition by antibodies from mice immunized with an oral V. cholerae vaccine (15 days post-vaccination). Figure 1A,C present the positions of these peptides and their respective measured intensities, as detected through chemiluminescent methods for mouse IgA and IgM antibodies present in the sera of mice that received the aOCV vaccination. Additionally, Figure 1B shows each peptide–epitope’s hierarchical position in terms of its reactivity, arranged from the top left to the bottom right position. Epitopes were defined based on the overlap of spots with signal strengths exceeding 30%, establishing the minimum sequence for these epitopes. The comprehensive list of the synthetic peptides and their positions on the membrane is in Appendix A. This analysis identified 18 IgG epitopes, with sizes ranging from 4 to 14 amino acids, which were generated because of vaccination with aOCV (Table 1).

### 2.2. Spatial Localization of the Major IgA and IgM Epitopes Within the Three Chain of the Toxin

The seven IgA and eleven IgM linear epitopes identified by SPOT synthesis analysis were distributed throughout TxA, TxB, and TxP (Table 1). The enterotoxin chain A contains two IgA epitopes (TxA-1A and TxA-2A) and all 11 IgM epitopes (TxA-1M-TxA-11. Enterotoxin B houses two IgA epitopes (TxA-3A and TxA-4A) and no IgM epitopes. Enterotoxin P has three IgA epitopes (TxB-5A to TxB-7A) and no IgM epitopes (Table 1).

### 2.3. Spatial Distribution of the Reactive Epitopes of Enterotoxin A, B, and P

The analysis of the seven IgA and eleven IgM epitopes, as determined by the SPOT synthesis approach, revealed their distribution throughout the V. cholera toxin proteins (Figure 2). The V. cholera protein gene can be divided into three distinct segments: a short signal peptide in the N-terminal extension (aa 1–18), the A1 subunit chain segment (aa 19–212), the A2 subunit chain segment (aa 213–258), the B subunit (aa 1–124), and the P protein (aa 1–221).

To pinpoint the location of these epitopes, we accessed the enterotoxin A, B, and P structures from the Protein Data Bank (PDB) or utilized AlphaFold DB predictions. Alternatively, we deduced the tertiary structure using the I-TASSER server (http://zhanggroup.org/I-TASSER/, accessed on 11 May 2023). Figure 2 showcases the resulting predicted structural models for the enterotoxin A, B, and P proteins. These models offer insights into the spatial localization of the eight reactive epitopes in CTA, three epitopes in CTB, and seven epitopes in CTP. Notably, most of the identified epitopes were situated within loop/coil structures, making them accessible to the solvent and located on the protein’s surface (Table 1 and Figure 2).

### 2.4. Specific and Cross-Immune IgA and IgM Epitopes

In exploring potential cross-immunity conferred by CT proteins, we employed CTA, CTB, and CTP sequences as templates in a multi-peptide matching search within the UniProtKB database. The criteria for this search involved the identification of peptides with four or more identical amino acids. Our analysis revealed that ten epitopes from *V. cholera* were specific (Table 1), while eight epitopes (TxA-1A, TxB-4A, TxP-5A, TxP-7A, TxA-1M, TxA-4M, TxA-5M, TxA-7M) exhibited cross-reactivity with E. coli or various bacteria. Specifically, the V. cholera-specific epitopes in CTA included TxA-2A, TxB-6A, TxA-2M, TxA-3M, TxA-6M, and TxA-8M to TxA-11M. Within CTP, only the epitope TxP-6A was identified as specific; in CTB, the specific epitope was TxB-3A. The epitopes TxP-5A and TxP-6A were determined to be exact within CTP (Table 1).

### 2.5. Reactivity of MAP4 and Chimeric Peptides via ELISA

It is well-known that serological assays based on bacterial proteins may exhibit cross-reactivity, underscoring the need to identify individual linear epitopes in the CT involved in the vaccination process. By restricting the antigens employed in an assay to capture reactive antibodies definitively specific to *V. cholera* through sequence analysis, developing a diagnostic ELISA that eliminates the potential for false positive results becomes feasible. To achieve this, three IgA and three IgM epitopes were carefully selected, synthesized, and subjected to ELISA analysis, both as individual peptides (Appendix A) and within multi-antigen peptides (MAP4) (Figure 3). A single IgA 45-mer and an IgM 45-mer chimeric peptide (Figure 3) encompassing the three chosen peptides of each subclass were synthesized. The sequences of the synthetic peptides are provided in Appendix A.

Upon examination, the sera from each immunized mouse showed reactivity to either the three synthetic peptides derived from IgA epitopes or the IgM epitopes, with a reassuring lack of response to the negative control peptide QEVRKYFCV (Vaccinia virus; Figure 1A, G9, and G17). This reactivity was detected 15 days after vaccination (Figure 3A) and 30 days later (Figure 3B), with no significant difference in antibody levels observed when using single peptides versus MAPs. Though the performance of the chimeric peptide IgA was maintained, considering the analysis of the sera collected 15 days after immunization against the 30 days after immunization, the performance of the chimeric IgM was slightly diminished. These results, as shown in Figure 3 and Appendix A, were found to be statistically significant (*p* < 0.001).

The choice of serum collection times was based on a clear understanding of the timeline of antibody development after infection. We know that IgM and IgA antibodies appear in the first week after infection or immunization, followed in the following weeks by IgG antibodies [37]. Infection occurs from a few hours to 5 days, and in most cases, it lasts 2 to 3 days. The period of transmissibility lasts as long as the vibrio is eliminated in the feces, which occurs, in most cases, until a few days after recovery. For surveillance purposes, the standard period accepted is 20 days. Therefore, in this study, we chose to collect serum from animals on the 15th and 30th days when vibriocidal, agglutinating, and protective antibodies are present [38,39,40].

Based on ROC curve analysis, the Area Under the Curve (AUC) for the epitope Tx45-IgA and Tx45-IgM was 0.9987 and 0.9896 (*p* < 0.0001), as detected by ELISA with a confidence interval of 95%, demonstrating high diagnostic accuracy for both the peptides. The Tx45-IgM peptide presented a lower reactivity than the Tx45-IgA, but both demonstrated 100% sensitivity and specificity.

## 3. Discussion

In this study, we identified linear B epitopes in CT recognized by IgA and IgM antibodies from mice vaccinated with an inactivated vaccine using peptide microarray analysis. CT serves as the primary virulence factor of *V. cholerae*, capable of inducing cholera symptoms with as little as 5 µg ingestion [41]. The serum reactivity against specific regions in CT demonstrated the vaccine’s high antigenic potential with a single dose. We identified two IgA and eleven IgM linear epitopes in CTA, two in CTB, and three in protein CTP, all accessible to the immune system due to their exposure on the molecular surface.

CTA is initially synthesized as a single polypeptide with a signal peptide (aa 1–18). The enzymatic activity begins after proteolytic cleavage, “nicking” at residue Arg-192, resulting in two functional domains: A1 (~21.8 kDa, aa 19–204) and A2 (~5.4 kDa, aa 206–258), covalently linked by a disulfide bond. The B subunit comprises five chains forming a pentameric ring around a central pore structure. CTA1 is responsible for ADP-ribosyltransferase activity, while the α-helical CTA2 anchors CTA1 and CTB5 together [42,43].

CTB, a crucial component of koCVs, is known to induce a robust protective response capable of neutralizing CT in the gut at low concentrations [9]. This response, mainly of the IgG type [9], with IgA and IgM responses [44] also playing important blocking roles, is acquired against CT during an infection or oral vaccination. Importantly, this protection is both T-dependent and crucial for developing a robust immune response against CT [44].

Our study demonstrated that the CTB subunit possesses only two major IgA epitopes among the seven detected. A previous study showed that when administered orally, two synthetic peptides of the TxB (aa 30–50; aa 50–75) induced a protective secretory IgA response in the gut [45]. Other authors demonstrated similar results in rats [46,47]. However, only the TxB- (aa 66–75) epitope identified in our study showed a structural correlation with the peptide (aa-50–75) described previously [45]. The aa 30–50 peptide from the previous study is possibly a minority epitope and was not identified in our studies. However, none of these circulating IgA epitopes shared IgM or IgG [33] sequences, underscoring the unique and distinct nature of the immunological response to TxB.

During the immune response, isotype change is a protective property that has the function of the organism producing antibodies with greater affinity to neutralize the antigen. After antigen recognition, recombination occurs between DNA segments of the genes that encode the constant regions of the heavy chains, determining the isotype [48]. The production time of these isotypes is variable and appears to depend on the antigen and the host. Consequently, some types of antibodies have a short half-life while others take a long time to disappear from circulation, which may explain the identification and not persistence of some isotypes of immunoglobulins/epitopes in the case of cholerae. Therefore, it is important to know which epitopes induce antibodies of longer duration and neutralizing capacity [48].

The IgA presents biological functions particularly dependent on its tertiary structure [49] and is given as a monomer (IgA1 circulating) or homodimer (IgA2 mucosal) after its synthesis [50] and different glycosylation profiles with IgA1 possessing more sialic acid than IgA2 [51]. The serum IgA can induce pro-inflammatory responses, such as releasing cytokines and chemokines, phagocytosis, degranulation, and formation of neutrophil extracellular traps (NETs) [52]. While IgA2 is composed of two IgA molecules (homodimer IgA), a joining protein (J chain), and a secretory component exert different biological functions in the mucosal tissues such as microbiota homeostasis, pro-inflammatory on neutrophils and macrophages, and opsonization of the bacteria [53] and even converts anti-inflammatory intestinal CD103+ DCs to a pro-inflammatory phenotype, which protects against invading pathogens, but might also result in chronic inflammation [54]. Independent of its biological functions, the heavy chains of the IgA immunoglobulins are produced in response to encountered antigen/specific epitopes independent of its subclasses [55,56,57,58]. Therefore, the analysis of circulating IgA, which is in most cases IgA1 [54], is sufficient to answer our question in advancing the qualitative identification of the major cholera epitope repertoire for diagnosis purposes and, in general, immune response.

In mouse models, it has been demonstrated that the primary immune response to cholerae is dominated by IgG against CTB, which is protective [58,59]. However, our study indicates that CTB also elicits IgA and IgM responses. Still, these responses appear weak compared to the number of identified IgG epitopes, despite the IgM being involved in opsonophagocytosis of the bacteria [34,60]. Importantly, it is a known fact that constant stimulation of the immune response leads to the production of long-lasting secretory IgA [61].

Each isotype is designed to tackle specific challenges, especially involving affinity. Normally, the IgM is the first antibody to emerge during embryonic development, and the humoral immune response is followed by IgA and IgG [62,63]. However, some IgM is also produced in secondary and subsequent responses, and after somatic hypermutation, although other isotypes dominate the later phases of the antibody response. The IgM can exist under distinct forms, including monomeric, membrane-bound IgM within the B cell receptor (BCR) complex, pentameric and hexameric IgM in serum can provide effective anti-microbial immunity [64,65], and secretory IgM on the mucosal surface. However, the regulation of humoral immune responses and B cell tolerance by the IgM Fc receptor (FcμR) is poorly understood [44,66].

Each isotype might prefer certain epitopes or residues of amino acids, which may be better suited to deal with specific types of invaders [67]. However, these epitopes possess different residues (right or left) but shared residues indicating distinct response preferences and evolution of the immune response, providing immunoglobulins with a higher binding or neutralization ability. Therefore, changes in the affinity will allow the immune system to be highly adaptable and effective in fine tuning, ready to handle diverse threats or situations [68].

Several IgG rapid antigen detection-based tests are available for cholerae diagnosis, such as immunochromatographic lateral flow devices like Crystal VC and Cholkit, which detect the presence of the O1 or O139 antigen in watery diarrheal stools and others [69,70,71]. However, limitations exist, and these tests may only sometimes offer high specificity [59]. Molecular testing using polymerase chain reaction (PCR) has also been used for diagnosis but primarily for research and surveillance [72,73]. Darkfield microscopy is another diagnostic approach for detecting *V. cholerae*, but it needs the required sensitivity for reliable diagnosis.

Thus, more precise, rapid diagnostic assays are needed. The IgA and IgM epitopes identified in this study can be applied in an ELISA assay to identify the primary immune response and provide insights into serological immunity. These unique IgA and IgM epitopes hold the potential for developing chimeric polyproteins for diagnostic purposes.

Vaccine production is typically a labor-intensive and costly experimental process. Nonetheless, these immunobiological molecules/agents are the most potent defenses against infectious diseases.

Presently, there are at least three pre-qualified oral cholera vaccines (OCVs): Dukoral^®^ Crucell, The Netherlands), Shanchol™ (Shantha, Biotechnics-Sanofi Pasteur, India), and Euvichol-Plus^®^ (EuBiologics Co., Seoul, Republic of Korea). While both categories have demonstrated protective effects against cholerae [4,74,75,76], the immunological mechanisms involving B-cell and/or T-cell immunity remain unclear [77,78]. It is well established that the key stimulator of innate immunity and subsequent adaptive immune responses is the LPS O antigen [6,10], which promotes long-term mucosal protection [8,13,79,80,81]. Hence, the pressing question is how to create more effective cholera vaccines, as current vaccine efficacies hover around 60% [1,27,82].

The findings on mice receiving a combination vaccination demonstrates a substantial increase in vibrocidal, IgM OSP-specific serum responses, and OSP-specific IgM memory B-cell responses. However, the research also indicates that enhanced opsonophagocytosis or antibody-dependent cytotoxic activity in the intestinal lumen does not seem to play a role in mediating protection against cholera, however, may present neutralizing ability. Even though cell-free antibody-based killing via complement lysis might be a possibility in the intestinal lumen, viability studies in animals have shown that bactericidal activity is not necessary for protection from disease [83,84,85,86]. These novel findings provide a deeper understanding of cholera protection mechanisms and vaccination responses, sparking curiosity in the field of microbiology and immunology.

Therefore, identifying immunodominant epitopes that interact with antibodies generated in response to vaccination can contribute to selecting better antigens and improve our understanding either for diagnostics or vaccination.

## 4. Materials and Methods

### 4.1. Immunization of Mice

We orally immunized thirty female BalbC mice, each weighing between 15 and 21 g, with 20 µL of the SchancolTM cholerae vaccine (lot SCN021A15; Shantha Biotechnics Ltd. in Muppiriddipalli, Telangana, India). Normally the infection occurs within a few hours to 5 days, with most cases lasting 2 to 3 days. The period of transmission lasts if the vibrio is eliminated in the feces, which occurs in most cases up to a few days after recovery. For surveillance purposes, the standard period accepted is 20 days. Fifteen days following vaccination, we collected blood samples and performed a complete bleed at the 30-day mark [34]. In parallel, blood samples were collected from 30 healthy, unvaccinated mice to serve as controls. The serum from each group was separately collected, divided into 0.5 mL aliquots in Eppendorf tubes, and stored at −20 °C. The study received ethical approval from the Ethics Commission of UNIGRANRIO (no 052/2021) study center ethics committee.

### 4.2. Synthesis of the Cellulose Membrane-Bound Peptide Array

Two peptide libraries comprising 122 peptides were synthesized to cover the full coding sequences of cholera enterotoxin A (P01555, 258 amino acids), enterotoxin B (P01556, 124 amino acids), and Toxin P (P29485, 221 amino acids) of serotype O1. The synthesis was performed on amino-PEG500-UC540 cellulose membranes (Intavis Bioanalytical Instruments, Köln, Germany) using standard SPOT synthesis protocols [87].

Each peptide was 15 amino acids long with a 10-amino-acid overlap, created with an Auto-Spot Robot ASP-222 (Intavis Bioanalytical Instruments AG, Köln, Germany). To enhance the libraries, a GSGSG spacer sequence was added at the amino and carboxy termini of each protein (positions A1, D5, D6, C4, C5, and F2). The libraries were constructed and programmed using the MultiPep 1 software (Intavis, Bioanalytical Instruments, Köln, Germany). Negative and positive control peptides were included: a negative control peptide (QEVRKYFCV from Vaccinia virus, located at spots G9 and G17) and positive controls such as KEVPALTAVETGATN (Poliovirus, at spots G3 and G11), GYPKDGNAFNNLDRI (Clostridium tetani, at spots G5 and G13), and YDYDVPDYAGYPYDV (Influenza hemagglutinin, at spots G7, G15, and G24).

The coupling reactions were followed by a blocking step using 4% acetic anhydride in *N*, *N*-dimethylformamide (DMF). The Fmoc protecting group was removed from the N-terminus of each peptide using 20% piperidine in DMF. This cycle of coupling, blocking, and deprotection was repeated for each amino acid addition. After the final amino acid was coupled, sidechain deprotection was performed with a solution of dichloromethane, trifluoroacetic acid, and triisopropylsilane (1:1:0.05, *v*/*v*/*v*), followed by ethanol washing. The synthetic peptide membranes were immediately prepared for probing

### 4.3. Screening of SPOT Membranes

The SPOT membranes were washed with TBS-T (50 mM Tris, 136 mM NaCl, 2 mM KCl, and 0.05% Tween^®^ 20, pH 7.4) for 10 min, then blocked in TBS-T containing 1.5% BSA for 90 min at 8 °C with agitation. Following additional washes with TBS-T, the membranes were incubated for 12 h with pooled sera from ten vaccinated mice, diluted 1:150 (for IgA or IgM detection) in TBS-T with 0.75% BSA. After thorough washing with TBS-T, the membranes were incubated for 1 h with goat anti-mouse IgM and IgA alkaline phosphatase-conjugated antibodies (Sigma Chemical Co., Saint Louis, MO, USA; diluted 1:5000) prepared in TBS-T with 0.75% BSA. Further washes with TBS-T and CBS (50 mM citrate-buffered saline) were performed. The chemiluminescent substrate Nitro-Block II Enhance (Applied Biosystems, Waltham, MA, USA) was added to finalize the reaction. The integrity of the library synthesis was validated by the reactivity of control peptides with human sera.

### 4.4. Scanning and Measurement of Spot Signal Intensities

Chemiluminescent signals were captured using an Odyssey FC imaging system (LI-COR Bioscience, Lincoln, NE, USA) with slight modifications to previously reported conditions [88]. A digital image was generated at 5 MP resolution, and signal intensities (SI) were analyzed using TotalLab TL100 software (v 2009, Nonlinear Dynamics, Newcastle-Upon-Tyne, UK). Background signals, determined from the negative control, were subtracted from the SI values, which were then normalized as a percentage of the highest signal detected. Epitopes were identified as sequences comprising two or three contiguous spots with normalized SI values of 30% or more. For epitopes involving three or more contiguous spots with SI values of 30% or greater, identification relied on visual evaluation.

A comparative analysis of the reactivity index of the normalized SPOTs was conducted using a dimensional hierarchical approach as previously described [89].

### 4.5. Preparation of Single and Multi-Antigen Peptides (MAPs)

The single and multi-antigen peptides were synthesized using a standard solid-phase synthesis protocol [90], employing Wang and tetrameric Fmoc2-Lys-B-Ala Wang resins (Appendix A). The constructs were generated using an automated peptide synthesizer (MultiPep-1, CEM Corp., Charlotte, NC, USA). Side chains of tetrafunctional Fmoc-protected amino acids were safeguarded with TFA-labile protecting groups where necessary. After assembling the peptide sequences, Fmoc groups were removed, and the peptide resin was cleaved and fully deprotected using a TFA/H_2_O/EDT/TIS mixture (94/2.5/2.5/1.0, *v*/*v*) for 90 min.

The peptides were precipitated by adding chilled diethyl ether, followed by centrifugation at 30,000× *g* for 10 min at 4 °C. The resulting pellet was dissolved in 10% aqueous AcOH (*v*/*v*), lyophilized, and stored as a dry powder. When required, MAPs were dissolved in water, centrifuged at 10,000× *g* for 60 min at 15 °C, and the supernatant was filtered through a Centricon™ filter (Merck Millipore, Burlington, MA, USA). Purification was conducted using an XBridge BEH C18 column (2.7 μm, 5 cm × 4.6 mm) integrated with a Waters AutoPurify HPLC system (Waters Corp., Newcastle, Australia). The flow rate was set at 1.2 mL/min, with mobile phase A consisting of 0.05% formic acid in water (18 MΩ × cm) and mobile phase B comprising 0.05% formic acid in acetonitrile. A gradient of 0–97% phase B over 40 min was employed, with detection at 200–300 nm using a diode array detector.

For ESI-TOF analysis, peptides were dissolved in deionized water to a final concentration of 10 µg/mL, and formic acid was added to achieve a 0.1% concentration. Mass spectrometry was performed using a Waters UPLC Acquity-I Class system (Waters Corp., Newcastle, Australia), with 1 µL/min sample injections. Ion detection spanned a range of 1000 to 11,500 *m*/*z*.

### 4.6. Preparation of 45-mer Chimeric Peptides

Two chimeric multiepitope peptides, Tx45-IgA and Tx45-IgM, were synthesized in sequence, incorporating two to six glycine spacers (Appendix A). The synthesis was performed using the Fmoc protocol on TentaGel-S-NH2 resin [91]. An automated peptide synthesizer (MultiPep-1, CEM Corp., Charlotte, NC, USA) was utilized, along with Fmoc-protected amino acids and appropriate protecting groups. Following synthesis, the peptides were cleaved from the resin, deprotected, and precipitated. Their identities were verified using mass spectrometry techniques, including MALDI-TOF or electrospray ionization.

### 4.7. In-House ELISA

Peptide-based ELISA (pELISA) assays were performed as described previously [91]. 4HB NUNC plates were coated with 80 ng of each peptide in a Na_2_CO_3_–NaHCO_3_ coating buffer (pH 9.6) overnight at 4 °C. After three PBS-T washes, the plates were blocked with PBS-T containing 2.5% BSA for 2 h at 37 °C. Diluted mouse sera (1:100) were applied and incubated for 2 h at 37 °C. The plates were washed and incubated with goat anti-mouse IgA-HRP or IgM-HRP antibodies for 2 h at 37 °C. Following further washing, pNPP substrate was added, and absorbance was measured at 405 nm. The serum dilution was optimized using a titration series and ROC curve analysis.

### 4.8. Structural Localization of IgG Epitopes and Bioinformatics Tools

To identify the locations of epitopes within the 3D molecular structures of the enterotoxin A, B, and P proteins from *V. cholerae*, in silico protein models were generated using the I-TASSER server (http://zhanglab.ccmb, accessed on 10 April 2023). The models were selected based on the highest C-score and TM-score (topological evaluation value) [92]. Additionally, the resulting 3D structural models were cross verified using the AlphaFold v3 database [93].

Sequence homologies for *V. cholerae* were analyzed by comparing the identified sequences with annotated proteins from other organisms available in the UniProt database (http://www.uniprot.org/, accessed on 12 April 2023) and through multiple peptide matching using the tool at https://research.bioinformatics.udel.edu/peptide match/index.jsp (accessed on 10 May 2023).

### 4.9. Statistical Analysis

The data were analyzed using the R program (version 3.6.0) and R Studio. To assess the statistical significance between the two samples, a paired *t*-test was employed, with significance considered at *p* ≤ 0.05. GraphPad Prism version 5.0 was used to analyze the receiver operating characteristic curve.

## 5. Conclusions

A resolutive assessment of the immune response was conducted using a peptide array that was directly generated on a cellulose membrane. This approach facilitated the identification of the major antigenic determinants in enterotoxins A, B, and P, which were recognized by antibodies from mice that had been orally vaccinated with a single dose of the aOCV. In total, we identified seven IgA and eleven IgM epitopes that were distributed throughout the bacterial CT protein. Specifically, six epitopes were in the A1 chain of the enterotoxin, two in the A2 chain, three in enterotoxin B, and seven in enterotoxin P. Notably, six of these epitopes exhibited a degree of similarity to proteins in other pathogens, indicating a high potential for cross-reactivity. Furthermore, we defined ten *V. cholera*-specific epitopes (three IgA and seven IgM) and elucidated their spatial locations within protein structural models. The efficacy of three of these IgA epitopes (TxA-2A, TxB-3A, and TxP-6A) and three IgM epitopes (TxA-2M, TxA-3M, and TxA-6M) in detecting antibodies produced in response to vaccinations in mice was confirmed using an indirect ELISA. This comprehensive molecular characterization of linear IgA and IgM epitopes in CT holds significant promise for developing poly epitope chimeric proteins as produced by our group for Chagas disease [87], which could be employed in next-generation rapid diagnostic tests. Additionally, our findings underscore the value of epitope mapping in enhancing our understanding of the IgM and IgA immune response to current and future vaccines, ultimately leading to improved production of specific neutralizing antibodies.

## Figures and Tables

**Figure 1 ijms-26-03507-f001:**
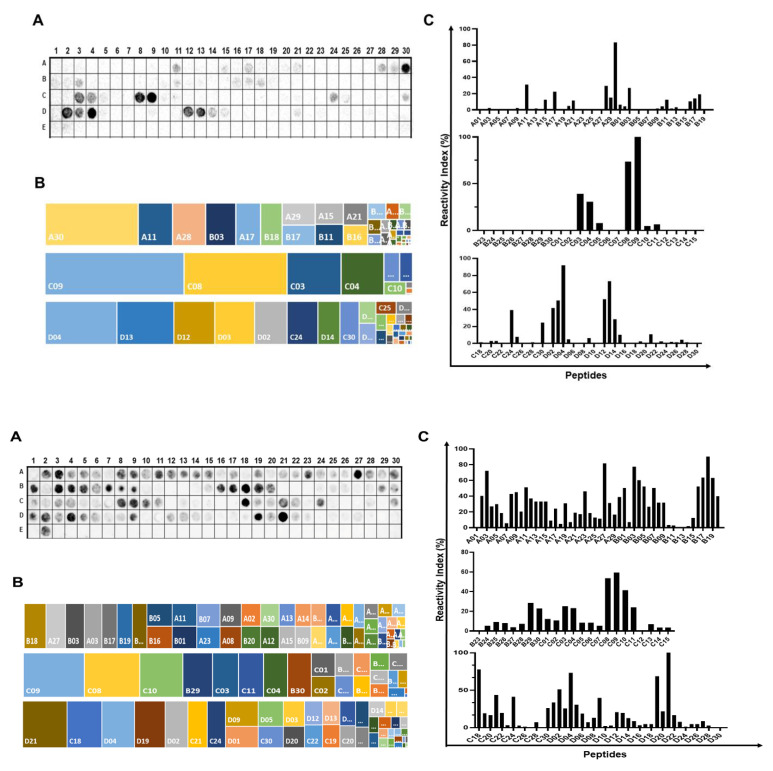
IgA (**upper**) and IgM (**low**) epitope mapping of *V. cholerae* toxin A (P01555), B (P01556), and P (P29485). A membrane-bound peptide library representing the three chains of the toxin was probed with a pool of vaccinated mouse sera (*n* = 15), and reactivity was detected using goat anti-mouse IgA (I) and IgM (II) alkaline phosphatase labeled secondary antibody and chemiluminescence substrate. Panels (**A**) present an image of the peptide array showing reactivity as dark circles. The panels in (**B**) show the hierarchy position of each positive peptide and, in (**C**), the percentage signal after normalizing the signals to the positive and negative controls. The sequences of peptides in each position are listed in Appendix A.

**Figure 2 ijms-26-03507-f002:**
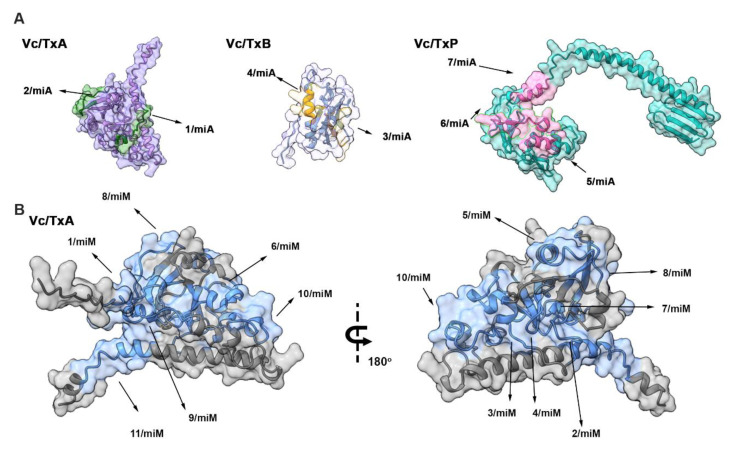
Epitope localization in three-dimensional structures of *V. cholerae* toxins. (**A**) IgA epitopes in *V. choleraee* toxins A (TxA), B (TxB), and P (TxP) epitopes are presented and colored within models constructed using the structure of toxin A, B, and P predicted by AlphaFold V3 (**B**) IgM epitope mapping within *V. choleraee* toxin A. Images were created using AlphaFold with ChimeraX (https://elearning.vib.be/courses/alphafold/lessons/vib-training-session-alphafold/topic/chimerax-display-alphafold-predictions-and-error-estimates-using-chimerax/).

**Figure 3 ijms-26-03507-f003:**
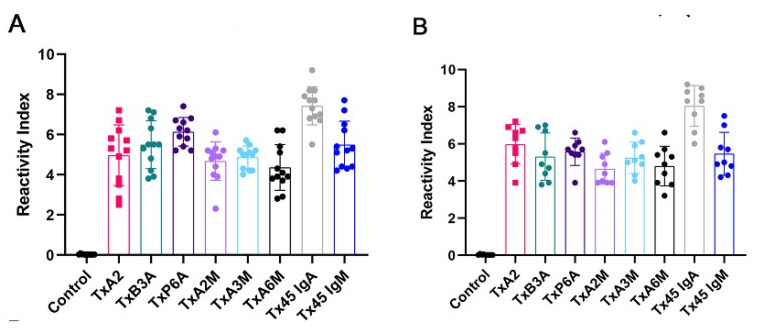
Reactivity of vaccinated mice (*n* = 12) sera 15 (**A**) and 30 (**B**) days post vaccination against the MAP4 peptides (TxA-2A, TxB-3A, TxP-6A, TxA-2M, TxA-3M, and TxA-6M) and chimeric tripeptides (Tx45-IgA and Tx45-IgM) by ELISA. As controls were used, sera of unvaccinated mice (*n* = 30). The relative index was calculated using the cut-off of each MAP to normalize the measured value from the sera of 12 vaccinated mice. The ROC curve was employed to establish the peptide cut-off values (Tx45-IgA and Tx45-IgM; 0.053 = 0.056), evaluate reactivity (99–100%), assess specificity (100%), and calculate the Area Under the Curve (AUC) values (ranging from 0.9987 to 0.9896).

**Table 1 ijms-26-03507-t001:** A list of B-cell linear IgA and IgM epitopes identified in TxA, TxB, and P was deduced from the overlap of consecutive positive peptides with signal intensities greater than 30%.

Protein Code	Code	aa	Sequence	Second Structure *	Peptide Search **
P01555	TxA-1A	51	RGTQMNINLYDHARG	C	*E. coli*
	TxA-2A	146–160	YRVHFGVLDEQLHRN	C	Sp
P01556	TxB-3A	66–75	REMAIITFKN	C + H	Sp
	TxB-4A	81–90	SQKKAIERMK	H	*E. coli*
P29485	TxP-5A	31–45	KPERLIGTPSIIQT	C + H	Sp
	TxP-6A	81–90	AIKRTRDFLN	C + H	Sp
	TxP-7A	126–135	QKKSVKERIK	C + H	Various bacteria
P01555	TxA-1M	11–25	FLSSFSYANDDKLYR	C	Various bacteria
	TxA-2M	41–50	MPRGQSEYFD	C	Sp
	TxA-3M	56–64	NINLYDHAR	C + H	Sp
	TxA-4M	71–80	VRHDDGYVST	C	*E. coli*
	TxA-5M	91–105	GQTILSGHSTYYIYV	C + H	Various bacteria
	TxA-6M	111–125	NMFNVNDVLGAYSPH	C	Sp
	TxA-7M	131–145	VSALGGIPYSQIYGW	C	Various bacteria
	TxA-8M	151–160	GVLDEQLHRN	C	Sp
	TxA-9M	171–185	RGYRDRYYSNLDIAP	C	Sp
	TxA-10M	191–205	GLAGFPPEHRAWREE	C	Sp
	TxA-11M	236–250	VKRQIFSGYQSDIDT	C + H	Sp

Sp, specific epitopes; C, coil; H, helix; * based on an I-TASSER analysis; ** UNIPROT.

## Data Availability

The data presented in this study are available on request from the corresponding author.

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
