# Peer review of "Dynamics of IgM and IgA Antibody Response Profile Against Vibrio cholerae Toxins A, B, and P"

_ijms, 2025, doi:10.3390/ijms26083507_

Round 1
Reviewer 1 Report
Comments and Suggestions for Authors
Cholera, caused by the bacterium Vibrio cholerae, remains a major public health problem, especially in areas with limited access to clean water and sanitation. The cholera toxin plays a critical role in the pathogenesis of the disease, and understanding its interaction with the immune system is crucial for the development of effective vaccines. In their previous work, the authors focused on mapping B-cell epitopes of Vibrio cholerae toxins, mainly targeting IgG responses. Building on this, the current study aims to identify and characterize the IgA and IgM epitopes of cholera toxins to gain both immunological and structural insights. By using techniques such as SPOT synthesis, the authors aim to further elucidate the immune recognition of cholera toxins, an important step towards improving vaccine strategies and disease control. The manuscript is well organized and the results are clearly presented. Given the importance of cholera, this research could be of great benefit in controlling the disease and promoting vaccination against cholera toxins. I recommend the manuscript for publication in the current form.
Author Response
Cholera, caused by the bacterium Vibrio cholerae, remains a major public health problem, especially in areas with limited access to clean water and sanitation. The cholera toxin plays a critical role in the pathogenesis of the disease, and understanding its interaction with the immune system is crucial for the development of effective vaccines. In their previous work, the authors focused on mapping B-cell epitopes of Vibrio cholerae toxins, mainly targeting IgG responses. Building on this, the current study aims to identify and characterize the IgA and IgM epitopes of cholera toxins to gain both immunological and structural insights. By using techniques such as SPOT synthesis, the authors aim to further elucidate the immune recognition of cholera toxins, an important step towards improving vaccine strategies and disease control. The manuscript is well organized, and the results are clearly presented. Given the importance of cholera, this research could be of great benefit in controlling the disease and promoting vaccination against cholera toxins. I recommend the manuscript for publication in the current form.
R: Thank you for taking the time to review the manuscript.

Reviewer 2 Report
Comments and Suggestions for Authors
De Simone et al. report on the identification of cholera toxin epitopes in synthetic peptides reactive with sera from mice immunized with an oral cholera vaccine. IgA and IgM antibody reactivity with the peptides was identified by ELISA. The peptide library synthesized mapped across the entire sequence of the cholera toxin protein. Knowledge of functional epitopes and functional antibodies will enhance diagnostic and vaccine development studies and result in more effective vaccines.
1. The labels in figure 1 are very difficult to read. The figure should be replaced.
2. Lines 280-282: “affinity (of) epitopes”?
3. Lines 208-215: “no significant difference in antibody levels” to the six individual peptides and the two 45 mer peptides reactive with mouse sera at 15 days and 30 days (Figure 3). Are peptides Tx45 containing IgA and IgM reactive epitopes immunogenic? Have they been used for immunization or only used for ELISA? Provide the rationale for bleed out at 30 days since duration of antibody response is an important component and consideration in determining how to maintain longer lasting immune response with cholera vaccines.
4. Were mouse antisera to oral cholera vaccine tested for titers, neutralization of toxin or other functional characteristics, in order to know the biological significance of the identified epitopes?
5. Clarification is required regarding the focus of the reported research, whether the peptide evaluation and epitope identification is for diagnostic platform design or vaccine design.
6. It appears that only serum IgA and IgM antibody reactivities to epitopes on the peptides were tested. There is no information provided for secretory antibody (for both IgA and IgM) reactive epitopes, or T epitopes. Are these known? See lines 229-230.
7. Mouse immunization: add a statement for IACUC approvals of animal studies.
8. Line 333: Indicate if mice used for immunization were male, female or mixed?
Author Response
De Simone et al. report on the identification of cholera toxin epitopes in synthetic peptides reactive with sera from mice immunized with an oral cholera vaccine. IgA and IgM antibody reactivity with the peptides was identified by ELISA. The peptide library synthesized mapped across the entire sequence of the cholera toxin protein. Knowledge of functional epitopes and functional antibodies will enhance diagnostic and vaccine development studies and result in more effective vaccines.
- The labels in Figure 1 are very difficult to read. The figure should be replaced.
R: Thank you, figure 1 was replaced.
Lines 280-282: “affinity (of) epitopes”?
R: Thank you, the word epitopes was removed.
- Lines 208-215: “no significant difference in antibody levels” for the six individual peptides and the two 45 mer peptides reactive with mouse sera at 15 and 30 days (Figure 3). Are Tx45 peptides containing IgA and IgM reactive epitopes immunogenic? Have they been used for immunization or only used for ELISA? Provide the rationale for bleeding out up to 30 days since the duration of antibody response is an important component and consideration in determining how to maintain a longer-lasting immune reaction with cholera vaccines.
R: The peptides were only used in vitro to validate the epitopes. A polypeptide containing only IgA epitopes (Tx45-IgA) and another peptide containing only IgM epitopes (Tx45-IgM (please see the legend of Figure 3 and line 207) were evaluated.
Serum collection times were based on a previous study in which we evaluated the IgM and IgG response of mice vaccinated with the vaccine used in the study. IgM peaked 15 days after vaccination and IgG 30 days later [33]. Therefore, collection was performed on the 15th and 30th day.
- Were mouse antisera to oral cholera vaccine tested for titers, neutralization of toxin, or other functional characteristics to know the biological significance of the identified epitopes?
R: Our study aimed to identify epitopes for diagnostic purposes only, not vaccines.
- Clarification is required regarding the focus of the reported research, whether the peptide evaluation and epitope identification is for diagnostic platform design or vaccine design.
R: Thank you, yes, for diagnostic purposes. The sentence was completed with our reference (please see lines 119-120 and 326-327).
- It appears that only serum IgA and IgM antibody reactivities to epitopes on the peptides were tested. No information is provided for secretory antibody (for both IgA and IgM) reactive or T epitopes. Are these known? See lines 229-230.
R: We did not evaluate the secretory IgA-IgM response, but a paragraph was added to the discussion.
- Mouse immunization: add a statement for IACUC approvals of animal studies.
R: This information is in lines 485-487. The study was approved by the UNIGRANRIO (CEUA 052/2021) study center ethics committee and conducted under good clinical practice and all applicable regulatory requirements.
- Line 333: Indicate if the mice used for immunization were male, female, or mixed.
R: Thank you, was inserted (line 333).

Round 2
Reviewer 2 Report
Comments and Suggestions for Authors
1. How was the level of IgA (Tx45-IgA) determined to fall within the 15 day and 30 day mark for bleeding out the mice, since IgM and IgG levels were previously examined? Even though IgM and IgA antibodies appear early, differences are observed for IgA depending on the antigen and whether secretory IgA or serum IgA levels (titers) are measured.
2. It is still unclear why secretory IgA was not evaluated. A brief explanation in the discussion section would clarify this.
3. Add a statement for IACUC approvals for animal studies in the section 5.1.Immunization of mice.
4. Line 208: "...recognized by each subclass of immunoglobulins..."
Author Response
Review 2’s comments
- How was the level of IgA (Tx45-IgA) determined to fall within the 15-day and 30-day mark for bleeding out the mice, since IgM and IgG levels were previously examined? Even though IgM and IgA antibodies appear early, differences are observed for IgA depending on the antigen and whether secretory IgA or serum IgA levels (titers) are measured.
R: This answer was already answered in item 2 of the previous comments. IgA level or titer was not measured for epitope identification studies. The IgA titer value or level has no importance in identifying qualitatively IgA epitopes. In mice, the induction of IgA against an antigen begins one week after exposure, in the same way as IgM (informed in the text). Therefore, we chose to bleed the animals 15 days later to identify the IgA and IgM epitopes. Serum collection times were based on a previous study in which we evaluated the IgM and IgG response of mice vaccinated with the vaccine used. IgM peaked 15 days after vaccination and IgG 30 days later [34]. Therefore, collection was performed on the 15th and 30th day.
- It is still unclear why secretory IgA was not evaluated. A brief explanation in the discussion section would clarify this.
R: A paragraph was introduced in the manuscript to explain the differences within the IgAs and why it is unnecessary in our experiments to evaluate the secretory IgA. We are not interested in studying local or circulating immunity and/or the function and neutralizing capacity of IgA in the vaccination and post-vaccination process. Our primary focus is identifying EPITOPES, a crucial aspect of our research. Since the Fab of secretory and circulating IgAs are identical, they adequately answer our manuscript questions. Please see lines 284-299 (text in yellow).
During the immune response, isotype change is a protective property that has the function of the organism producing antibodies with greater affinity to neutralize the antigen. After antigen recognition, recombination occurs between DNA segments of the genes that encode the constant regions of the heavy chains, determining the isotype [48]. The production time of these isotypes is variable and appears to depend on the antigen and the host. Consequently, some types of antibodies have a short half-life while others take a long time to disappear from circulation, which may explain the identification and not persistence of some isotypes of immunoglobulins/epitopes in the case of cholerae. Therefore, it is important to know which epitopes induce antibodies of longer duration and neutralizing capacity [48].
“The IgA presents biological functions particularly dependent on its tertiary structure [49]. Are given as a monomer (IgA1 circulating) or homodimer (IgA2 mucosal) after its synthesis [50] and different glycosylation profiles with IgA1 possessing more sialic acid than IgA2 [51]. The serum IgA can induce pro-inflammatory responses, such as releasing cytokines and chemokines, phagocytosis, degranulation, and formation of neutrophil extracellular traps (NETs) [52]. While IgA2 is composed of two IgA molecules (homodimer IgA), a joining protein (J chain), and a secretory component exert different biological functions in the mucosal tissues such as microbiota homeostasis, pro-inflammatory on neutrophils and macrophages, and opsonization of the bacteria [53] and even converts anti-inflammatory intestinal CD103+ DCs to a pro-inflammatory phenotype, which protects against invading pathogens, but might also result in chronic inflammation [54]. Independent of its biological functions, the heavy chains of the IgA immunoglobulins are produced in response to encountered antigen/specific epitopes independent of its IgA1 or IgA2 subclasses [55-58] (The affinity change due to recognition of side amino acids of the epitope, but the central region of the epitope is the same). Therefore, the analysis of circulating IgA, which is in most cases IgA1 [54], is sufficient to answer our question in advancing the qualitative identification of the major cholera epitope repertoire for diagnosis purposes and, in general, immune response.
- Add a statement for IACUC approvals for animal studies in section 5.1.Immunization of mice.
R: This information was in lines 485-487. The study was approved by the UNIGRANRIO (CEUA 052/2021) study center ethics committee and conducted under good clinical practice and all applicable regulatory requirements. We have also inserted it in section 5.1 to emphasize the importance of ethical considerations in our research.
- Line 208: "...recognized by each subclass of immunoglobulins..."
R: This sentence was changed in the previous version to provide a more accurate and clear description of the recognition process by each subclass of immunoglobulins.

Round 3
Reviewer 2 Report
Comments and Suggestions for Authors
Text provided in the reviewer responses should be inserted into the body of the manuscript to help clarify inconsistencies and provide missing information regarding the antibodies.
A view of the revised manuscript shows that text is missing from the revised manuscript.
Comments on the Quality of English LanguageThe manuscript needs to be checked for English grammar and phrasing.
Author Response
- How was the level of IgA (Tx45-IgA) determined to fall within the 15-day and 30-day mark for bleeding out the mice, since IgM and IgG levels were previously examined? Even though IgM and IgA antibodies appear early, differences are observed for IgA depending on the antigen and whether secretory IgA or serum IgA levels (titers) are measured.
R: IgA titration was not previously performed during the 30 days after vaccination. Since it is known that IgM begins to be produced 4-5 days after contact with the antigen, IgA follows approximately this time in different other models, and IgG starts to increase in serum after one week, we decided to standardize 15 days to identify epitopes IgA and IgM and 30 days for IgG. However, at any of these times, less or more of all classes of immunoglobulins may be found. Antibody titration in our study is unimportant because the objective is to identify the majority epitopes. Therefore, we used a cut of 30% to consider the sequences identified as epitopes. Those detected above this value are shown in Table 1. This does not mean that there may still be minority epitopes that are below 30%. These epitopes normally do not have great significance due to the low circulating antibodies. The chimeric peptide antigen Tx45-IgA was used to confirm that by conjugating the three different epitopes, it is possible to have a higher detection level in the ELISA test than using simple peptides and to show that the antibody levels against these epitopes are maintained high throughout either in 15 and 30 days analyzed and therefore are good targets for the development of diagnostic theses.
- It is still unclear why secretory IgA was not evaluated. A brief explanation in the discussion section would clarify this.
R: This analysis was not included in the initial project submitted to CEUA. Since an experiment of this type requires many animals to have a sufficient sample quantity of secretion to perform Spot Synthesis, we chose to analyze only circulating IgA. A sentence was introduced justifying the absence of the experiment.
- Add a statement for IACUC approvals for animal studies in section 5.1. Immunization of mice.
R: OK, the information was inserted in the manuscript and was annexed to the e-mail.
- Line 208: "...recognized by each subclass of immunoglobulins..."
R: The sentence was corrected (line 208), and a new sentence was introduced (Lines 215-217). “Though the performance of the chimeric peptide IgA was maintained, considering the analysis of the sera collected 15 days after immunization against the 30 days after immunization, the performance of the chimeric IgM was slightly diminished.”